# Adenosine Metabolism in the Cerebral Cortex from Several Mice Models during Aging

**DOI:** 10.3390/ijms21197300

**Published:** 2020-10-02

**Authors:** Alejandro Sánchez-Melgar, José Luis Albasanz, Mercè Pallàs, Mairena Martín

**Affiliations:** 1Department of Inorganic, Organic and Biochemistry, Faculty of Chemical and Technological Sciences, Universidad de Castilla-La Mancha, School of Medicine of Ciudad Real, Regional Center of Biomedical Research (CRIB), 13071 Ciudad Real, Spain; alejandro.sanchez@uclm.es (A.S.-M.); mairena.martin@uclm.es (M.M.); 2Department of Pharmacology and Therapeutic Chemistry, Faculty of Pharmacy and Food Sciences, Institute of Neuroscience, University of Barcelona, 08028 Barcelona, Spain; pallas@ub.edu

**Keywords:** aging, adenosine metabolism, glutamate, animal models, purinergic signaling

## Abstract

Adenosine is a neuromodulator that has been involved in aging and neurodegenerative diseases as Alzheimer’s disease (AD). In the present work, we analyzed the possible modulation of purine metabolites, 5’nucleotidase (5′NT) and adenosine deaminase (ADA) activities, and adenosine monophosphate (AMP)-activated protein kinase (AMPK) and its phosphorylated form during aging in the cerebral cortex. Three murine models were used: senescence-accelerated mouse-resistant 1 (SAMR1, normal senescence), senescence-accelerated mouse-prone 8 (SAMP8, a model of AD), and the wild-type C57BL/6J (model of aging) mice strains. Glutamate and excitatory amino acid transporter 2 (EAAT2) levels were also measured in these animals. HPLC, Western blotting, and enzymatic activity evaluation were performed to this aim. 5′-Nucleotidase (5′NT) activity was decreased at six months and recovered at 12 months in SAMP8 while opposite effects were observed in SAMR1 at the same age, and no changes in C57BL/6J mice. ADA activity significantly decreased from 3 to 12 months in the SAMR1 mice strain, while a significant decrease from 6 to 12 months was observed in the SAMP8 mice strain. Regarding purine metabolites, xanthine and guanosine levels were increased at six months in SAMR1 without significant differences in SAMP8 mice. In C57BL/6J mice, inosine and xanthine were increased, while adenosine decreased, from 4 to 24 months. The AMPK level was decreased at six months in SAMP8 without significant changes nor in SAMR1 or C57BL/6J strains. Glutamate and EAAT2 levels were also modulated during aging. Our data show a different modulation of adenosine metabolism participants in the cerebral cortex of these animal models. Interestingly, the main differences between SAMR1 and SAMP8 mice were found at six months of age, SAMP8 being the most affected strain. As SAMP8 is an AD model, results suggest that adenosinergic metabolism is involved in the neurodegeneration of AD.

## 1. Introduction

Adenosine is a key regulator of the neurotransmission in the central nervous system operating through four receptors: A_1_, A_2A_, A_2B_, and A_3_. These receptors belong to the G-protein coupled receptors (GPCRs) family, and activate (A_2A_, A_2B_) or inhibit (A_1_, A_3_) adenylyl cyclase enzyme, thus modulating intracellular cAMP levels. Adenosine receptors’ expression and distribution seem to be region- and cell-type-specific in the brain [1,2,3]. Many studies have revealed that adenosine receptors, mainly A_1_ and A_2A_, are strongly altered in neurodegenerative [4,5,6,7,8] and neuropsychiatric disorders [9,10,11].

Adenosine and related metabolites, together with their converting enzymes (Scheme 1), have also been analyzed in several healthy and pathological conditions, including Alzheimer’s disease (AD) [12,13,14,15]. Furthermore, 5′-nucleotidase (5′NT) catalyzes the formation of adenosine from adenosine monophosphate (AMP) [16], and adenosine deaminase is involved in the degradation of adenosine to inosine [17].

AD is characterized by the extracellular deposition of amyloid β-peptide (Aβ) plaques and the intracellular formation of neurofibrillary tangles in the cerebral cortex [18]. In addition, age affects the brain independently of AD [19], and there is a growing number of proponents of the view that it is important to understand age-related changes to better understand AD. Therefore, studying the brain regions that are vulnerable to both aging and AD, such as the cerebral cortex, may allow us to disentangle disease-specific mechanisms from normal age-related events [20,21].

Age is the main risk factor for prevalent diseases, including neurodegenerative disorders (reviewed in [22]). A causal connection between reactive oxygen species, aging, age-related pathologies, and cell senescence is intensely studied [23]. Aging refers to biological change with time across the lifespan; however, senescence refers to normal progressive functional impairments that occur with age. Age-related disease refers to health conditions for which the incidence increases with age, such as cancer, heart disease, arthritis, osteoporosis, and cognitive decline (e.g., AD) [24].

Mice are extensively used in aging and senescence research due to their genetic and physiological similarity to humans [24]. Wild-type C57BL/6J is the most widely used inbred strain and is frequently used in studies related to aging and neurodegenerative disorders [25,26,27]. Senescence-accelerated mouse-prone 8 (SAMP8) is an ideal model to study AD, displaying age-related learning and memory disorders [28,29,30,31,32]. Senescence-accelerated mouse-resistant 1 (SAMR1) is usually considered as the control strain for SAMP8 mice studies, along with accelerated senescence but without learning and memory impairments [28,29].

The relationship between first-occurring AD-like changes and the monthly age in the brain of SAMP8 mice has been reported [28]. In brief, at three months of age, these mice have aberrant gene expression, oxidative stress, and tau hyperphosphorylation. At six months, they develop cognitive impairment, glial degeneration, inflammation, and Aβ deposition. Finally, at 12 months of age, they show alterations in dendrites, synapses, neurons, Aβ plaques, and neuron loss. The lifespan of SAMP8 mice is about 10–12 months, while SAMR1 mice have a lifespan of 19–21 months [29,33]. Regarding C57BL/6J, its lifespan is about 26–28 months [34,35]. As C57BL/6J mice exhibit a longer lifespan than SAMR1 and SAMP8 mice, we selected mice that were 4 and 24 months of age and approximately at the same survival point as seen for the 3- and 12-month-old SAM mice [33,36]. In fact, it has been reported that one-year-old SAMP8 mice showed age-associated characteristics in tissue histology, proliferation, and differentiation, which were comparable to two-year-old mice of the C57BL/6 strain [37].

Interestingly, age-related changes in adenosinergic signaling, including adenosine receptors levels, have been reported in these models. In relation to this, our group previously described a reduced expression of A_1_ associated with age [38,39], while increasing the density of A_2A_ was found as an early event in the whole brain from SAMP8 mice when compared to the SAMR1 strain [39].

Several GPCRs that can couple to the Gi/o subfamily are modified during aging. Among these receptors, adenosine A_1_ receptors have a reduced turnover rate, lesser functional output, and reduced protein levels (reviewed in [40]). In contrast, adenosine A_2A_ receptors, which are coupled to the Gs subfamily, seem to be potentiated by aging, and their overfunction accelerates neurodegeneration processes [41,42].

Homeostatic functions, neuromodulation, and neuroprotection via adenosine receptors also decrease with age [13,40,43]. In addition, aging and neurodegenerative diseases, such as AD, are generally associated with reduced cAMP signaling in the brain compared to normal levels (reviewed in [44]).

The potential role of dietary restriction in the prevention of phenotypic features during aging in the brain and its diverse mechanisms has been reviewed elsewhere [45]. In line with this, nutrient sensors, in particular AMP-activated protein kinase (AMPK) and sirtuins, tend to downregulate with age, and their pharmacological activation is reputed to increase longevity [46,47]. The AMPK is a serine/threonine kinase with a key role as an energy sensor in a wide variety of tissues. It coordinates cell growth, autophagy, and metabolism [48,49,50,51], and appears to be involved in the pathophysiology of AD [52]. AMPK is a trimer, with α-, β-, and γ-subunits. The α-subunit contains the kinase domain and its Thr^172^ residue is phosphorylated by an upstream kinase and determines its regulation. The binding of AMP, and to a lesser extent ADP, to the γ-subunit stimulates AMPK activity. Thus, changes in the ATP-to-ADP or ATP-to-AMP ratio leads to the allosteric activation of AMPK [53]. Therefore, enzymes involved in the modulation of AMP levels due to the conversion of AMP to adenosine (i.e., 5′NT) or adenosine to AMP (i.e., adenosine kinase), in turn, could control AMPK activation. Extracellular adenosine activates AMPK [54]; however, the authors of [55] reported the PKA-mediated inhibition of AMPK via the increased inhibitory phosphorylation of AMPK^Ser173^ and reduced activating phosphorylation of AMPK^Thr172^.

Adenosine signaling also interacts with many other neurotransmitter systems such as glutamatergic signaling [56,57], which is also modulated by aging and has a role in neurodegeneration [58,59,60,61]. The glutamatergic system has been found to be altered in the human brain in AD patients [62,63] and also in the whole brain of SAMP8 mice [64].

Therefore, the aim of the present work was to assess adenosine, related metabolites and enzymes (5′NT, ADA), and their potential implication on AMPK activation in the cerebral cortex. SAMR1 (senescence, normal impairment), SAMP8 (senescence, abnormal impairment as AD), and the wild-type C57BL/6J (aging, change in functionality with age) mice were analyzed during aging to decipher possible differences, if any, between these murine models. Glutamate and excitatory amino acid transporter 2 (EAAT2) levels were also measured and analyzed due to their relationship with adenosinergic signaling.

## 2. Results

### 2.1. Modulation of 5′-Nucleotidase Activity during Aging in the Cerebral Cortex from SAM and C57BL/6J Mice

Activities of 5′NT located in the plasma membrane and cytosolic fraction were measured separately. In the SAMR1 mice strain, 5′NT activity significantly increased from three to six months, in both the plasma membrane and cytosolic fractions, and returned at 12 months to similar levels measured than in the three-month-old mice (Figure 1). In SAMP8 mice, a reduction of 5′NT activity in both the plasma membrane and cytosolic fractions was found at six months vs. three months. At 12 months, these activities recovered to values similar to those measured at three months. When comparing SAMR1 and SAMP8 values, significantly different activities were found at three and six months in the plasma membrane and at six months in the cytosolic fraction. In the plasma membrane fraction, two-way ANOVA (strain × age) revealed a significant interaction between factors (*F_(2, 15)_* = 9.360, *p* = 0.0023), a nonsignificant main effect of the strain (*F_(1, 15)_* = 0.6574, *p* = 0.4302), and a nonsignificant effect of age (*F_(2, 15)_* = 1.248, *p* = 0.3154). In the cytosolic fraction, two-way ANOVA revealed a significant interaction between factors (*F_(2, 14)_* = 5.461, *p* = 0.0176), a significant main effect of the strain (*F_(1, 14)_* = 12.29, *p* = 0.0035), and a nonsignificant effect of age (*F_(2, 14)_* = 0.09649, *p* = 0.9086). In C57BL/6J mice, 5′NT slightly, but not significantly, increased from 4 to 24 months in the plasma membrane fraction while remaining unchanged in the cytosolic fraction (Figure 2). Therefore, it seems that a different and opposite modulation process takes place in SAM mice strains during aging.

### 2.2. Modulation of Adenosine Deaminase Activity during Aging in the Cerebral Cortex from SAM and C57BL/6J Mice

Adenosine deaminase (ADA) activity located in the plasma membrane fraction was measured. ADA activity significantly decreased from 3 to 12 months in the SAMR1 mice strain, while a significant decrease from 6 to 12 months was observed in the SAMP8 mice strain (Figure 3a). Two-way ANOVA (strain × age) revealed a significant main effect of age (*F_(2, 13)_* = 6.821, *p* = 0.0094), a nonsignificant main effect of the strain (*F_(1, 13)_* = 0.02968, *p* = 0.8659), and a nonsignificant interaction between factors (*F_(2, 13)_* = 1.403, *p* = 0.2808). Regarding the C57BL/6J mice strain, no significant differences were detected from 4- to 24-month-old mice.

### 2.3. Modulation of Adenosine and Related Metabolite Levels during Aging in the Cerebral Cortex from SAM and C57BL/6J Mice

Adenosine and related metabolite levels were quantified by HPLC. In SAMR1 mice, adenosine levels remained unchanged during aging (Figure 4a). Inosine, xanthine, hypoxanthine, and guanosine levels showed a similar profile, increasing from three to six months and decreasing from 6 to 12 months (Figure 4b–e). These changes from three to six months and from 6 to 12 months were statistically significant for xanthine and guanosine levels (Figure 4c,e). In SAMP8 mice, there was a more erratic profile during aging for all the measured metabolites, and no significant differences were detected (Figure 4). When comparing SAMR1 and SAMP8 values, only hypoxanthine and guanosine levels were significantly different at six months (Figure 4d,e). Two-way ANOVA revealed no significant changes in adenosine levels with age (*F_(2, 15)_* = 0.6746, *p* = 0.5242), strain (*F_(1, 15)_* = 1.250, *p* = 0.2811), or age × strain (*F_(2, 15)_* = 0.3222, *p* = 0.7294). For inosine levels, two-way ANOVA revealed no significant changes with age (*F_(2, 14)_* = 0.5776, *p* = 0.5741) and age × strain (*F_(2, 14)_* = 0.4688, *p* = 0.6352), and significant changes with strain (*F_(1, 14)_* = 5.423, *p* = 0.0354). For xanthine levels, two-way ANOVA revealed no significant changes with strain (*F_(1, 15)_* = 0.1877, *p* = 0.6710) and age × strain (*F_(2, 15)_* = 0.2410, *p* = 0.7889), and significant changes with age (*F_(2, 15)_* = 3.916, *p* = 0.0428). For hypoxanthine levels, two-way ANOVA revealed no significant changes with strain (*F_(1, 14)_* = 2.974, *p* = 0.1066), age (*F_(2, 14)_* = 0.03323, *p* = 0.9674), and age × strain (*F_(2, 14)_* = 3.698, *p* = 0.0514). For guanosine levels, two-way ANOVA revealed no significant changes with strain (*F_(1, 14)_* = 2.334, *p* = 0.1488) and age × strain (*F_(2, 14)_* = 2.264, *p* = 0.1406), and significant changes with age (F_(2, 14)_ = 5.752, P = 0.0150). In C57BL/6J, adenosine levels were significantly decreased by age, whereas increased levels of inosine and xanthine were observed in older mice (Figure 5). However, hypoxanthine and guanosine levels remained unchanged from 4 to 24 months.

### 2.4. AMPK Activation State during Aging in the Cerebral Cortex from SAM and C57BL/6J Mice

AMPK and its phosphorylated state (p-AMPK) were analyzed by Western blotting. In SAMR1 mice, AMPK levels seem to be lower at 6 and 12 months than at three months; however, this decrease was not statistically significant (Figure 6a). In SAMP8 mice, a significant decrease was detected from three to six months, followed by an also significant recovery of AMPK levels from 6 to 12 months (Figure 6b). In C57BL/6J mice, AMPK levels were similar at 4 and 24 months (Figure 6c). Finally, the activation of AMPK, measured as the p-AMPK/AMPK ratio, remained unchanged during aging in all analyzed mice (Figure 6a–c).

### 2.5. Glutamate and Excitatory Amino Acid Transporter 2 Levels during Aging in the Cerebral Cortex from SAM and C57BL/6J Mice

We measured the glutamate level in the cytosolic fraction of the cerebral cortex. In SAMR1 mice, glutamate levels were similar at three and six months, but decreased at 12 months. In SAMP8 mice, glutamate levels diminished from three to six months, and significantly increased from 6 to 12 months (Figure 7). While glutamate levels are similar in both SAM strains at three months, they were significantly different at 6 and 12 months. Interestingly, SAMP8 had lower glutamate levels than SAMR1 at six months, while the contrary was found at 12 months. Two-way ANOVA (strain × age) revealed a nonsignificant main effect of age (*F_(2, 15)_* = 1.016, *p* = 0.3857) and strain (*F_(1, 15)_* = 0.1543, *p* = 0.7000), and a significant interaction between factors (*F_(2, 15)_* = 7.357, *p* = 0.0059). In C57BL/6J mice, there is an increase in glutamate levels from 4 to 24 months (Figure 8).

Finally, we measured the protein levels of excitatory amino acid transporter 2 (EAAT2), which is responsible for the vast majority of glutamate clearance, in the plasma membrane fraction of the cerebral cortex. In SAMR1 mice, EAAT2 levels were similar at three and six months, but significantly increased from 6 to 12 months (Figure 9a). In SAMP8, EAAT2 levels significantly increased from three to six months, and decreased from 6 to 12 months (Figure 9b). In C57BL/6J mice, EAAT2 protein levels were similar at 4 and 24 months (Figure 9c).

## 3. Discussion

As summarized in Scheme 1, adenosine monophosphate (AMP) can be degraded to adenosine by ecto-5′-nucleotidase (ecto-5′NT), and adenosine can be converted via inosine and hypoxanthine to xanthine by adenosine deaminase (ADA), purine nucleoside phosphorylase (PNP), and xanthine oxidase (XO), respectively. Guanosine monophosphate (GMP) can be metabolized to guanosine, which can be converted via guanine and hypoxanthine to xanthine by 5′NT, PNP, guanase, and XO, respectively. Hypoxanthine-guanine phosphoribosyltransferase (HGPRT) is a purine salvage enzyme, which converts hypoxanthine and guanine to IMP and GMP, respectively [65].

There is evidence of age-dependent changes in the activity of some nucleoside metabolism enzymes, which could result in different nucleoside levels. Age profoundly affects 5′NT activity in different areas of the Fischer 344 rat brain of young, intermediate, and old rats (2, 12, and 24 months, respectively). The activity of cytosolic- and ecto-5′-nucleotidase increased in the cerebral cortex of old- and intermediate-age rats when compared to young animals [15]. Our results in SAMR1 mice indicate that 5′NT activity increased from three to six months. However, in SAMP8 mice, a reduction in 5′NT activity in both the plasma membrane and cytosolic fractions at six months vs. three months, which is in agreement with the decrease we previously found in the whole brain of SAMP8 mice from five to seven months of age [38]. Concerning C57BL/6J mice, a slight increase from 4 to 24 months was found in plasma membrane-associated 5′NT activity.

Levels of inosine and adenosine increased with age in samples from the frontal cortex of old men as compared to their middle-aged counterparts [66]. Thus, an increase in 5′NT activity with age likely leads to an increase in adenosine levels in the brain [15]. However, we did not find any modulation of adenosine levels in SAMR1 mice despite the increased 5′NT activity from three to six months of age. Even decreased 5′NT activity from three to six months, as detected in SAMP8 mice, was unable to modulate adenosine levels. Interestingly, detected 5′NT activities (plasma membrane and cytosolic) and guanosine levels were lower in SAMP8 mice than in SAMR1 mice. Moreover, the increased guanosine levels detected in SAMR1 mice at six months versus three months could be, at least in part, a reflection of the similarly increased 5′NT activity measured at this age. However, the positive relationship between 5′NT and guanosine values is apparently lost in SAMP8 mice. We must note that the total content of nucleosides measured here may not represent the amount available at the synaptic cleft; however, it provides an overview of the nucleoside metabolism in the cerebral cortex of these animals. A more complex scenario is envisaged from the reduced adenosine levels found in C57BL/6J mice, where no significant changes in 5′NT activity were detected. Guanosine levels also remained unchanged with age in these mice, thus reinforcing the relationship between 5′NT and guanosine values. Therefore, adenosine-metabolizing enzymes, such as ADA, could also be regulated by age and contribute to the modulation of adenosine and inosine levels. We found a slight increase in ADA activity from 4 to 24 months in the C57BL/6J mice strain, which could have contributed to the lower adenosine and higher inosine levels detected in old mice. In turn, ADA activity progressively decreased in SAMR1 mice throughout the aging period, while in SAMP8, mice this decrease was observed from 6 to 12 months. In line with this, ADA activities were reported higher in young than in adult Wistar rats [67]; even age-associated alterations of ADA activity in different brain regions have been reported [15]. Interestingly, adenosine, related metabolites, and their converting enzymes are altered in several cortical areas from the postmortem human brain of AD patients, even at the early stages of the disease [12].

Several age-related changes in a wide range of behaviors have been reported in C57BL/6J mice from young to old age (8, 47, 73, and 99 weeks of age). Among these changes, reduced working and spatial memory, alongside impaired cued fear memory, were observed in old-aged mice compared to those in young C57BL/6J mice [68]. A significant spatial learning and memory decline was also detected between 3- and 15-month-old C57BL/6J mice [69]. In line with this, it has been reported that a calorie-restricted diet may improve the hippocampus-dependent spatial learning ability of C57BL mice via an increase in AMPK expression [70]. Total AMPK levels in the brain were not significantly different between old (16–18 months) and young (9–12 weeks) C57BL/6 mice, although the basal AMPK phosphorylation level was higher in old animals [71]. Moreover, no differences in total AMPK content have been reported in the prefrontal cortex from 22- to 33-week-old C57BL/6J mice [72]. In agreement, in the present study, total AMPK and phosphorylated AMPK (p-AMPK) levels in the cerebral cortex from C57BL/6J mice were preserved from 4 to 24 months.

Regarding SAM strains, the activity of AMPK in the cerebral cortex of young (two-month-old) SAMP8 mice was reported significantly higher than that of SAMR1 controls, thus inhibiting tau hyperphosphorylation [73]. These authors suggested that AMPK activation at an early stage (i.e., presymptomatic two-month-old SAMP8 mice) without prominent AD pathologies plays a protective role against detrimental stress in the cerebral cortex, as the activation of cortical AMPK inhibits the GSK3β-mediated hyperphosphorylation of tau. In addition, activated AMPK (p-AMPK) accumulated in pre-tangle-bearing and tangle-bearing neurons in postmortem brain samples in AD and other major tauopathies [74]. We have not directly compared the levels and activity of AMPK between SAMP8 and SAMR1 mice, as these samples were electrophoresed in different gels. However, it is interesting to note that at the age of six months, total AMPK levels significantly decrease in SAMP8 mice, and significant differences (e.g., 5′NT, guanosine, hypoxanthine, glutamate) between these two SAM strains are detected at this age. This would point to a loss of the cited AMPK-mediated compensatory mechanism at the age of six months. Age-related changes in the AMPK function have been reviewed both at the cell and organism levels. Unfortunately, further investigation is needed to accurately determine how aging modulates the activity and expression of AMPK, mainly in mammals [46].

An age-related reduction in glutamate levels in the whole brain of SAMP8 mice from five- to seven-month-old mice was previously reported by our group [64]. In the present work, covering a wider time frame (3, 6, and 12 months) and a limited region (cerebral cortex), glutamate levels in SAMR1 mice were similar at three and six months, but decreased at 12 months. In SAMP8, glutamate levels diminished from three to six months, and significantly increased from 6 to 12 months. Finally, increased glutamate levels were also detected in aged C57BL/6J mice.

Glutamate levels will depend on its synthesis, uptake, and/or metabolic fate, with a relative contribution that may be different during aging.

Regarding glutamate synthesis, glucose provides the carbon for the de novo synthesis of major neurotransmitters (e.g., glutamate) and the glucose metabolism provides the energy to sustain neurotransmission (reviewed in [75]). A generally increased metabolism of glucose during aging (one to eight months) has been described for SAMR1 and SAMP8 mice strains in different brain areas including the cerebral cortex [76,77]. However, when compared to SAMR1, glucose metabolism in the SAMP8 brain was normal at one month of age but decreased from two to three months of age and onwards (eight months) [76]. Lower glucose metabolism in SAMP8 than in SAMR1 at the age of 7–8 months old was also reported, and the extent of impaired glucose metabolism in the cerebral cortex of SAMP8 mice correlated significantly with the severity of the learning impairments observed in these mice [78]. In agreement, lower glutamate synthesis from glucose and acetate (only metabolized by astrocytes) and lower synthesis of glutamine from acetate have been reported in eight-month-old SAMP8 compared to two-month-old mice [79], suggesting that the glutamate metabolism in the brains of eight-month-old SAMP8 mice is altered [80]. In older (40 weeks) SAMP8 mice, significantly increased serum fasting glucose (1.66-fold) levels were observed compared with young (12 weeks) SAMP8 mice. Nevertheless, there was no difference in the fasting glucose levels of both SAMR1 and C57BL/6 mice strains between 12 and 40 weeks [81]. At 12 months of age, the expression levels of GLUT1 and GLUT3 protein, the main glucose transporters in the brain, were decreased in the cerebral cortex of both SAMR1 and SAMP8 mice when compared to the corresponding four-month-old animals [82].

Regarding glutamate uptake, the neuroprotective capacity of astrocytes by clearing extracellular glutamate has been analyzed in primary cultures enriched in astrocytes using cerebral cortical tissue from two-day-old SAMP8 and SAMR1 mice. Interestingly, glutamate uptake in SAMP8 astrocyte cultures was significantly less than in SAMR1 cultures, while the expression of the glutamate–aspartate transporter (GLAST) was not different between SAMP8 and SAMR1 astrocytes. Moreover, the neuroprotective capacity of astrocytes, evaluated in cocultures with cortical neurons, was lower in SAMP8 astrocytes [83]. In addition, astrocytes aged in vitro have a reduced ability to maintain neuronal survival [84]. This reduction in glial glutamate uptake capacity in SAMP8 may elevate extracellular glutamate levels, leading to neuronal excitotoxicity [85], and contribute to premature learning and memory deficits observed not only in this murine model of early aging but also of AD. In fact, astrocyte senescence and its putative role in the pathologic progress of AD has been recently reviewed [86]. Astrocytes cultured from neonatal SAMP8 mice present similar alterations to those described in the whole brains of SAMP8 mice at 1–5 months of age [83]. Thus, it is conceivable that astrocytes (mal)function could contribute to the significantly higher glutamate levels that we detected in 12-month-old SAMP8 mice when compared with the SAMR1 strain.

In line with this, we detected an increase in EAAT2 levels in SAMR1 mice from 6 to 12 months, when glutamate levels are lower. Furthermore, in SAMP8 mice, the highest EAAT2 protein level is detected at six months, when glutamate content is lowest, suggesting the importance of glutamate uptake in the final glutamate level found in the cerebral cortex of SAM strains.

Another contributing factor to glutamate levels could be a reduced glutamate uptake into synaptic vesicles. In fact, the protein expression of vesicular glutamate transporter (VGLUT) isoforms 1–3 and synaptophysin was decreased in an age-dependent manner in the cerebral cortex of SAMP8 mice of 2, 6, and 12 months, which could indicate that the glutamatergic synaptic transmission was weakened in the brain of aging SAMP8 [87].

Therefore, glutamate synthesis from glucose can be the main, but not unique, contributing factor to glutamate levels at six months of age, while glutamate uptake processes would be more important at 12 months in both SAMR1 and SAMP8 mice.

Glutamic acid decarboxylase (EC 4.1.1.15; GAD) promotes the transformation of excessive glutamate to γ-aminobutyric acid (GABA). Interestingly, GAD activity was lower in the cortex from older (24 months) C57BL/6J mice compared to younger mice (four months) [88]. Lower GAD activity could be a primary contributor to the higher glutamate content in 24-month-old compared to four-month-old C57BL/6J mice that we reported here, as we found no changes in EAAT2 protein levels (i.e., glutamate uptake) between 4 and 24 months.

An additional point of complexity is that changes in brain glutamate levels seem to be region-specific. Thus, the basal glutamate content was increased in CA1, and remained unchanged in CA3 and DG, leading to elevated hippocampal glutamate from 3 to 11 months of age in C57BL/6J mice [89]. In SAMR1 and SAMP8 mice, glutamate monitoring between 2 and 14 months of age revealed a different age-related modulation in the cerebral cortex and the hippocampus [90]. Interestingly, these authors reported that the high K^+^-evoked release of glutamate was age-dependently decreased in slices of SAMR1, and similarly in SAMP8, where it was also decreased at early stages (two and six months of age); however, it was increased at 9 and 11 months of age.

Together, these results potentially highlight six-month-old SAMP8 mice as the most affected among the three strains analyzed in this work. SAMP8 mice constitute both a senescence-accelerated model and a model for studying the initial neurodegenerative alterations in AD [30,31,32,91,92]. In fact, impairments in spatial learning and increased oxidative stress can be detected in SAMP8 mice as early as three months old, followed by the impairment in spatial memory and increased tau hyperphosphorylation at five months, hippocampal cognitive deficits at six months, and gliosis and increased levels of soluble Aβ at eight months [28,31]. The results presented herein regarding changes in several components of adenosinergic signaling, and the modulation of adenosine receptors by age in both SAMR1 and SAMP8 mice [38,39], are in agreement with the implication of the adenosinergic system in AD neuropathology [93].

Adenosine receptors, mainly A_1_ and A_2A_, are modulated in the brain during the neurodegenerative process. Hippocampus-dependent learning and memory were more altered by caffeine exposure during preadolescence (42 days old) and adolescence (180 days old) compared with adulthood in C57BL/6J mice, probably due to age-related differences in adenosine receptor levels [94]. The densities of adenosine A_1_ and A_2A_ receptors changed oppositely in areas such as the cerebral cortex and hippocampus during aging. Thus, A_1_ receptor binding to the cerebral cortex in old (24 months) Wistar rats was a 40% of that detected in young (six weeks old) rats, while A_2A_ receptor binding was two-fold higher in old than in young rats [41]. Our group previously described that, in the whole brain of SAMR1 mice, adenosine A_1_ and A_2A_ receptors were decreased and increased, respectively, from three weeks to six months of age [39]. In SAMP8 mice, we reported that A_1_ protein levels were maintained from three weeks to six months [39] and decreased from five to seven months of age [38], while A_2A_ protein levels remained unaltered between three weeks and six months [39] and between five and seven months of age [38]. Interestingly, the increased density of the A_2A_ receptor was found as an early event in the whole brain from SAMP8 mice when compared to the SAMR1 strain [39]. In AD patients, A_1_ and A_2A_ were found to be increased in the frontal cortex [4], whereas a higher A_2A_ density [95,96] and lower A_1_ [97,98] has been described in the hippocampus. Since adenosine A_1_ receptors are mainly related to neuroprotection [14], their loss in very young SAMP8 mice suggests the involvement of A_1_ receptors in the pathogenesis of age-associated neurodegenerative diseases [39]. Moreover, the overfunction of A_2A_ receptors accelerates the neurodegeneration process [14,42], and the apparently preserved levels of A_2A_ receptors in SAMP8 mice during aging would assure some pathological events; for instance, an overstimulated glutamate release [10,99]. Furthermore, a key regulator of A_2A_-receptor-mediated signaling seems to be 5′NT (reviewed in [100]). The colocation and physical association of 5′NT and A_2A_ receptors has been reported within the same membrane microdomains; for instance, in synaptosomes from the striatum of C57BL/6 animals [101,102]. Thus, extracellularly generated adenosine would immediately activate A_2A_ receptors, which in turn would promote glutamate release. Interestingly, the temporal profile of glutamate content in SAMP8 and C57BL/6J mice reported here was paralleled to plasma 5′-NT activity during aging.

## 4. Materials and Methods

### 4.1. Animals

Cerebral cortexes from 33 male mice were used in this work: 13 SAMR1 (3 months, *n* = 5; 6 months, *n* = 5; 12 months, *n* = 3), 9 SAMP8 (3 months, *n* = 3; 6 months, *n* = 3; 12 months, *n* = 3), and 11 C57BL/6J mice (4 months, *n* = 7; 24 months, *n* = 4). Animals received a standard diet (2018 Teklad Global 18% Protein Rodent Maintenance Diet, Harlan). All the mice had food and water ad libitum and were kept in standard conditions of temperature (22 ± 2 °C) and 12:12 h light–dark cycles (300 lux/0 lux). All procedures were performed in accordance with the institutional guidelines for the care and use of laboratory animals established by the Ethical Committee for Animal Experimentation at the University of Barcelona (670/14/8102, approved at 11/14/2014).

### 4.2. Plasma Membrane Isolation

Plasma membranes of the cerebral cortex from SAM and C57BL/6J mice were isolated as previously described [103]. Samples were homogenized in 20 volumes of isolation buffer (50 mM Tris-HCl, pH 7.4, containing 10 mM MgCl_2_ and protease inhibitors) in a Dounce homogenizer (10xA, 10xB). After homogenization, samples were centrifuged for 5 min at 1000× *g* in a Beckman JA 21 for 30 min at 27,000× *g*. The resulting supernatant was considered as the cytoplasmic fraction, and the pellet was resuspended in an isolation buffer and considered the plasma membrane fraction. Samples were stored at −80 °C until needed. Protein levels were quantified by the Lowry method using bovine serum albumin as standard.

### 4.3. 5’-Nucleotidase Activity Assay

5′-Nucleotidase activity was measured as previously reported [104]. Briefly, 20 μg of protein were preincubated at 37 °C for 10 min in the reaction medium (50 mM Tris-HCl, 5 mM MgCl_2_, pH 9). Then, the reaction was initiated by adding AMP at a final concentration of 500 μM and stopped 20 min later by adding 10% trichloroacetic acid. The samples were chilled on ice for 10 min and then centrifuged at 12,000× *g* for 4 min at 4 °C. The supernatants were used to measure inorganic phosphate released using KH_2_PO_4_ as Pi standard. The nonenzymatic hydrolysis of AMP was corrected by adding samples after trichloroacetic acid. Incubation times and protein concentration were selected to ensure the linearity of the reactions. All samples were run in duplicate. Enzymatic activity is expressed as nanomolar Pi released/min mg protein.

### 4.4. Adenosine and Related Metabolite Levels Quantification by HPLC

Chromatographic analysis was performed with Ultimate 3000 U-HPLC (Thermo Fisher, Madrid, Spain) and data peaks were processed with Chromeleon 7 (Thermo Fisher, Madrid, Spain) as previously described [12]. HPLC diode array was used working at a 254 nm wavelength. Purine standards and samples (40 μL) were injected in a C18 column of 4.6 mm × 250 mm with a 5 μm particle size. Two solvents were used for gradient elution: Solvent A—20 mM phosphate buffer solution (pH 5,7), and Solvent B—100% methanol. The gradient was 95% (11 min), 80% (9 min), and 95% (2 min) in Solvent A. The total run time was 22 min with a constant flow rate of 0.8 mL/min at 25 °C. Retention times for hypoxanthine, xanthine, inosine, guanosine, and adenosine were 3.5, 3.9, 8.4, 9.4, and 15.5 min, respectively. Purine levels were interpolated from the standard curve constructed with five concentrations of each purine ranging from 0.1 to 500 μM. Data were then normalized to the protein concentration of each sample.

### 4.5. Immunodetection of p-AMPK and AMPK by Western Blotting

In total, 20 μg of protein of cytosolic fraction from each sample were mixed with loading buffer containing 0,125 M Tris (pH 6.8), 20% glycerol, 10% β-mercaptoethanol, 4% SDS, and 0.002% bromophenol blue, and heated at 50 °C for 5 min. Protein was electrophoresed on 10% SDS–PAGE gel using a miniprotean system (Bio-Rad, Madrid, Spain) with molecular weight standards (Bio-Rad). Protein transfer to nitrocellulose membranes was carried out in the iBlot^TM^ Dry Blotting System (Invitrogen, Madrid, Spain). Membranes were washed with PBS-Tween 20, blocked with PBS containing 5% skimmed milk, and then incubated with the primary antibodies at 4 °C overnight at 1:1000 dilution for anti-EAAT2 (Abcam, ab41621), 1:600 for anti-AMPK (Abcam, ab207442), 1:1000 for antiphosphorylated AMPK (Abcam, ab23875), and 1:2000 for anti-GAPDH (Abcam, ab8245). GAPDH was used as a gel loading control. After rinsing, the membranes were incubated with the corresponding secondary antibody (Bio-Rad, GAMPO 170-6516, GARPO 172-1019) at a dilution of 1:5000 in PBS containing 5% skimmed milk for 1 h. The antigen was visualized using the ECL chemiluminescence detection kit (Amersham, Madrid, Spain) in a G:Box chamber, and specific bands were quantified by densitometry using GeneTools software (Syngene).

### 4.6. Glutamate Level Quantification

Glutamate level was quantified as indicated in the manufacturer’s protocol (Molecular Probes, Ref. A12221). Briefly, 50 μL of diluted samples were mixed into 96-well black plates with 50 μL of the reaction mix containing Amplex Red, Horseradish peroxidase, L-alanine, and L-glutamate-pyruvate transaminase and L-glutamate oxidase. Fluorescence was measured in kinetic mode for 30 min. Data were then interpolated to a standard curve and normalized to the amount of protein. Excitation/Emission was detected at Ex/Em = 530/590 nm.

### 4.7. Adenosine Deaminase Activity Assay

Adenosine deaminase (ADA) activity was measured with an enzyme activity assay kit (Abcam ab204695) according to the manufacturer’s protocol (Abcam, Cambridge, UK). The plasma membrane fraction was diluted 1:100 in ADA Buffer Assay in a 96-well plate and assayed in duplicate. Then, the 96-well plate was read at Ex/Em = 535/587 nm as a kinetic curve for 30 min. Sample values were obtained by interpolation in an inosine standard curve performed in parallel on the same plate as previously described [12].

### 4.8. Statistical and Data Analysis

Data are mean ± SEM. One-way ANOVA and two-way ANOVA were performed in SAM mice (3-, 6-, and 12-month groups), and Student’s *t*-test was performed in C57BL/6J mice (4- and 24-month groups) for statistical analysis, as indicated in the aforementioned figure captions. The overall comparisons between the SAMR1 and SAMP8 mice were tested with two-way ANOVA. The main effects tested were strain (measured parameter differs between the SAMR1 and SAMP8 mice), age (measured parameter changes with age), and interaction between strain and age (age-related changes are different between the SAMR1 and SAMP8 mice). Differences between mean values were considered statistically significant at *p* < 0.05. GraphPad Prism 6.0 program was used for statistical and data analysis (GraphPad Software, San Diego, CA, USA).

## 5. Conclusions

Our data indicate that adenosine metabolism is differently altered in the cerebral cortex of SAMR1, SAMP8, and the wild-type C57BL/6J mice. SAMP8 strain was the most affected, mainly at the age of six months, which is in line with cognitive decline and other features of AD pathology previously described in this strain. Since adenosine has been implicated in multiple key biological processes in the CNS such as neuroinflammation, homeostasis control, neurotransmitter release (e.g., glutamate), among others, the enzymes involved in adenosine metabolism, with a particular emphasis in 5′NT, should be considered as a promising target for neurodegenerative diseases, particularly AD.

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
