# Peer review of "Adenosine Metabolism in the Cerebral Cortex from Several Mice Models during Aging"

_ijms, 2020, doi:10.3390/ijms21197300_

Round 1
Reviewer 1 Report
The study from Sanchez-Melgar is an interesting manuscript evaluating adenosine and metabolites, and glutamate levels and enzymes activities related to adenosine metabolism in the cortex of different mice strains during aging. However, the results description needs improvement and it was not mentioned the different age between SAM and C57Bl strains. Additionally, the discussion also needs improvement, as mentioned bellow.
Minor:
Abstract:
-Please define the abbreviation AD at its first mention.
- Please add “mice” before “strains” (4th line).
Major:
Abstract: - Information regarding the differences among mice strains must be added in the abstract.
Results:
- Figure 1 and 2: It was not clear the reason why the mice strains SAMR1 and SAMP8 were analyzed at 3, 6, and 12 months and compared to C57BL at 4 and 24 months. Please clarify it in the results section.
- Figure 3: the description of the results does not follow exactly what is observed in the graphs. For instance, in figure 3b there is no significant effect for inosine levels. However, at page 4, lines 116-117 it is erroneously indicated. And, in the figure 3e the description to SAMP8 at 12 months is also not correct (or, there is a mistake and the authors mean SAMR1 mice). Fig 3 and 4: Again, the comparison with 4 and 24 months C57BL mice is not clear. The use of C57BL mice with different age from the SAM mice must be clearly justified, or these data can be supressed from the manuscript.
- Figure 5: the absence on pAMPK detection in SAMP8 mice may be mentioned in the results description (fig 5b).
- Discussion section: if the aim of the manuscript was doing a comparison among rodent strains, it will be interesting to specify the data obtained to each strain. For instance , page 8 first paragraph: which rats? Sprague-Dawley, Wistar, others?
- In the second paragraph (lines 195-206) the authors discuss the relationship between 5´-nucleotidase and adenosine levels. Perhaps they should also include guanosine levels in such discussion, as it is also formed by 5´NT activity. And, they might consider the total content of nucleosides may not represent the amount available at synaptic cleft.
- In the following paragraph (from line 211), the discussion is confusing. It is mentioned that pAMPK levels are higher in older mice (reference 59). How old are these mice? 24 months, as in the present study? And, these data from [59] are compared with the absence of pAMPk and AMPK levels in the present study. It is not clear what the authors are aiming to inform. Please reorganize this paragraph.
- page 9, paragraph from 232-251: Although the data are interesting, there are several conceptual errors in this paragraph. I agree that the increase in glutamate levels may be due to increased synthesis. However, GAD is not the enzyme responsible for glutamate synthesis. Is there any information regarding glucose uptake and metabolism (most of glutamate in the brain comes from glucose oxidation)? Or glutamine synthetase activity, that is responsible for recycling the glutamate taken up to astrocytes?
However, the idea that reduced glutamate uptake may be responsible for increasing glutamate levels in the cortical tissue is strange. The authors mean reduce vesicular glutamate uptake? The reduced VGluT1 cited from [64] may be related to a reduction in glutamatergic terminals. And, the authors must consider that the main responsible for glutamate uptake are the cell membrane EAATs.
- In the paragraph regarding adenosine receptors, again it is not clear whether the reduced expression of A1 and increased of A2A were observed at the same time points from the present study. And, the last phrase is puzzling: what is the relationship among 5´-NT and A2AR interaction and glutamate release? The suggestion that A2AR may have a role on regulating glutamate uptake and release it is known (Ciruela et al., 2006; Matos et al., 2013; 2015). This discussion should be reorganized.
Methods:
- It is not clear the exact number of mice used to each experiment. Please clarify the N to each group.
Conclusions:
- The conclusions are also not very clear. And the following phrase is not correct: “Notably, 5’NT activity similarly behaved as glutamate levels in SAMP8 mice … “
It seems the authors are considering that 5´-NT activity decrease in 12 months would be responsible for increasing glutamate levels? It seems to me that the relationship between 5´- NT and A2aR localization and interaction (proteins expression) may not be explained by 5´-NT activity.
Author Response
Reviewer: 1
The study from Sanchez-Melgar is an interesting manuscript evaluating adenosine and metabolites, and glutamate levels and enzymes activities related to adenosine metabolism in the cortex of different mice strains during aging. However, the results description needs improvement and it was not mentioned the different age between SAM and C57Bl strains. Additionally, the discussion also needs improvement, as mentioned bellow.
Minor:
Abstract:
-Please define the abbreviation AD at its first mention.
We have defined AD in the Abstract.
- Please add “mice” before “strains” (4th line).
We have added “mice” before “strains”.
Major:
Abstract: - Information regarding the differences among mice strains must be added in the abstract.
We have modified the Abstract to define the strains analyzed, and to include new results regarding ADA activity and EAAT2 levels.
Results:
- Figure 1 and 2: It was not clear the reason why the mice strains SAMR1 and SAMP8 were analyzed at 3, 6, and 12 months and compared to C57BL at 4 and 24 months. Please clarify it in the results section.
We have included in the Introduction section the rationale for selecting these mice strains of the analyzed ages. See Line 67: “Mice are extensively used in aging and senescence research due to their genetic and physiological similarity to humans [24]. Wild-type C57BL/6J is the most widely used inbred strain and they are frequently used in studies related to aging and neurodegenerative disorders [25-27]. Senescence-accelerated mouse-prone 8 (SAMP8) is an ideal model to study AD, displaying age-related learning and memory disorders [28-32], and senescence-accelerated mouse-resistant 1 (SAMR1) is usually considered as the control strain for SAMP8 mice studies, with also accelerated senescence but without learning and memory impairments [28,29].
The relationship between first occurred AD-like changes and monthly age in the brain of SAMP8 mice has been reported [28]. In brief, at 3 months of age, these mice have aberrant gene expression, oxidative stress and Tau hyperphosphorylation. At 6 months, they develop cognitive impairment, glial degeneration, inflammation, and Aβ deposition. Finally, at 12 months of age, they show alterations in dendrites, synapses, neurons, and Aβ plaques and neuron loss. The lifespan of SAMP8 mice is about 10–12 months, while SAMR1 mice is about 19–21 months [29,33]. Regarding C57BL/6J, its lifespan is about 26–28 months [34,35]. As C57BL/6J mice exhibit a longer lifespan than SAMR1 and SAMP8 mice, we selected 4 and 24 months of age, approximately at the same survival point as seen for 3 and 12 month-old SAM mice [33,36]. In fact, it has been reported that SAMP8 mice at 1-year-old showed age-associated characteristics in tissue histology, proliferation and differentiation, which were comparable to 2-year-old mice of the C57BL/6 strain [37].”
- Figure 3: the description of the results does not follow exactly what is observed in the graphs. For instance, in figure 3b there is no significant effect for inosine levels. However, at page 4,lines 116-117 it is erroneously indicated. And, in the figure 3e the description to SAMP8 at 12 months is also not correct (or, there is a mistake and the authors mean SAMR1 mice). Fig 3 and 4: Again, the comparison with 4 and 24 months C57BL mice is not clear. The use of C57BL mice with different age from the SAM mice must be clearly justified, or these data can be supressed from the manuscript.
We have corrected the description of results included in Figure 3 (now Figure 4) as follows (see Line 164): “Adenosine and related-metabolites levels were quantified by HPLC. In SAMR1 mice, adenosine levels remained unchanged during aging (Figure 4a). Inosine, xanthine, hypoxanthine and guanosine levels showed a similar profile, increasing from 3 to 6 months and decreasing from 6 to 12 months (Figure 4b, c, d, e). These changes from 3 to 6 months and from 6 to 12 months were statistically significant for xanthine and guanosine levels (Figure 4c, e). In SAMP8 mice, there was a more erratic profile during aging for all the measured metabolites, and no significant differences were detected (Figure 4). When comparing SAMR1 and SAMP8 values, only hypoxanthine and guanosine levels were significantly different at 6 months (Figure 4d, e).”.
We have justified the use of C57BL/6J at 4 and 24 months in the Introduction section (see Line 79): “Regarding C57BL/6J, its lifespan is about 26–28 months [34,35]. As C57BL/6J mice exhibit a longer lifespan than SAMR1 and SAMP8 mice, we selected 4 and 24 months of age, approximately at the same survival point as seen for 3 and 12 month-old SAM mice [33,36]. In fact, it has been reported that SAMP8 mice at 1-year-old showed age- associated characteristics in tissue histology, proliferation and differentiation, which were comparable to 2-year-old mice of the C57BL/6 strain [37].”.
-
Figure 5: the absence on pAMPK detection in SAMP8 mice may be mentioned in the results description (fig 5b).
p-AMPK detection and analysis were already included in the previous Figure 5b (now Figure 6b) in the first version of the manuscript. Nevertheless, we have modified Figure caption to mention AMPK and p-AMPK levels. See Line 199: “Figure 6. AMPK and p-AMPK levels during aging in cerebral cortex from SAM and C57BL76J mice. Cytosolic fractions of cerebral cortex from SAMR1 (a), SAMP8 (b), and C57BL/6J (c) mice were used to analyze the amount of AMPK and its activated state (p-AMPK). Western blots were performed as described in Methods. GAPDH was used as gel loading control. Data are mean ± SEM of three to seven different animals. One-way ANOVA revealed *** p < 0.001 significantly different from the 3 months value; & p < 0.05 significantly different from indicated value at 6 months.”.
-
Discussion section: if the aim of the manuscript was doing a comparison among rodent strains, it will be interesting to specify the data obtained to each strain. For instance , page 8 first paragraph: which rats? Sprague-Dawley, Wistar, others?
We have now specified in the Discussion section the rodent strains where the cited results were obtained.
-
In the second paragraph (lines 195-206) the authors discuss the relationship between 5´- nucleotidase and adenosine levels. Perhaps they should also include guanosine levels in such discussion, as it is also formed by 5´NT activity. And, they might consider the total content of nucleosides may not represent the amount available at synaptic
We have included, as suggested, the relationship between guanosine a 5’- nucleotidase, which is particularly interesting in SAMR1 mice. Therefore, we have included the following paragraph (see Line 262): “Interestingly, detected 5’NT activities (plasma membrane and cytosolic) and guanosine levels were lower in SAMP8 than in SAMR1 mice. Moreover, the increased guanosine levels detected in 6 months versus 3 months SAMR1 mice could be, at least in part, a reflect of the also increased 5’NT activity measured at this age. However, this positive relationship between 5’NT and guanosine values is apparently lost in SAMP8 mice.”
We agree that reported nucleoside levels may not represent the synaptic cleft content. We have stated this in the discussion section (see Line 266): “We must note that the total content of nucleosides measured here may not represent the amount available at synaptic cleft, but gives an overview of the nucleoside metabolism in the cerebral cortex of these animals.”.
-
In the following paragraph (from line 211), the discussion is confusing. It is mentioned that pAMPK levels are higher in older mice (reference 59). How old are these mice? 24 months, as in the present study? And, these data from [59] are compared with the absence of pAMPk and AMPK levels in the present study. It is not clear what the authors are aiming to inform. Please reorganize this
We agree these sentences can be a bit confusing. We measured both p-AMPK and AMPK levels in C57BL/6J, SAMR1 and SAMP8 strains. As Figure 3 shows in the first version of the manuscript (now Figure 4), there are no changes with aging. This absence of changes on AMPK levels agree with data from [59] and [60]. We have rewritten these sentences as follows (see Line 285): “In line with this, it has been reported that caloric restriction diet may improve hippocampus-dependent spatial learning ability of C57BL mice via an increase in AMPK expression [70]. Total AMPK levels in brain were not significantly different between old (16-18 months) and young (9-12 weeks) C57BL/6 mice, although basal AMPK phosphorylation level was higher in old animals [71]. Moreover, no differences in total AMPK content has been also reported in the prefrontal cortex from 22 and 33 weeks old C57BL/6J mice [72]. In agreement, in the present study total AMPK and phosphorylated AMPK (p-AMPK) levels in the cerebral cortex from C57BL/6J mice are preserved from 4 to 24 months.”.
In addition, we have also included some modifications in the following paragraph (see Line 300): “We have not directly compared the levels and activity of AMPK between SAMP8 and SAMR1 mice, as these samples were electrophoresed in different gels. However, it is interesting to note that at the age of 6 months total AMPK levels significantly decrease in SAMP8 mice, and the most differences (e.g. 5’NT, guanosine, hypoxanthine, glutamate) between these two SAM strains are detected at this age. This would point to a loss of the cited AMPK-mediated compensatory mechanism at the age of 6 months.”.
-
page 9, paragraph from 232-251: Although the data are interesting, there are several conceptual errors in this paragraph. I agree that the increase in glutamate levels may be due to increased synthesis. However, GAD is not the enzyme responsible for glutamate synthesis. Is there any information regarding glucose uptake and metabolism (most of glutamate in the brain comes from glucose oxidation)? Or glutamine synthetase activity, that is responsible for recycling the glutamate taken up to astrocytes?
We agree this section was confusing and incomplete. Therefore, we have rewritten this part concerning glutamate levels, and additional references about glucose metabolism and glutamate synthesis, release and uptake have been included. See Line 315: “Glutamate levels will depend on its synthesis, uptake and/or metabolic fate, with a relative contribution that may be different during aging.
Regarding glutamate synthesis, glucose provides the carbon for de novo synthesis of major neurotransmitters (e.g. glutamate) and glucose metabolism provides the energy to sustain neurotransmission (reviewed in [75]). A generally increased metabolism of glucose during aging (1 to 8 months) has been described for SAMR1 and SAMP8 mice strains in different brain areas including cerebral cortex [76,77]. However, when compared to SAMR1, glucose metabolism in the SAMP8 brain was normal at 1 month of age but decreased from 2-3 months of age onwards (8 months) [76]. Lower glucose metabolism in SAMP8 than in SAMR1 at the age of 7-8 months old was also reported, and the extent of impaired glucose metabolism in the cerebral cortex of SAMP8 mice correlated significantly with the severity of the learning impairments observed in these mice [78]. In agreement, lower glutamate synthesis from glucose and acetate (only metabolized by astrocytes), and lower synthesis of glutamine from acetate have been reported in 8-month-old SAMP8 compared to 2-month-old mice [79], suggesting that glutamate metabolism in brains of 8-month-old SAMP8 mice is altered [80]. In older (40 weeks) SAMP8 mice a significantly increased serum fasting glucose (1.66-fold) levels were observed compared with young (12 weeks) SAMP8 mice. Nevertheless, there was no difference in fasting glucose levels of both SAMR1 and C57BL/6 mice strains between 12 and 40 weeks [81]. At 12 months of age, the expression levels of GLUT1 and GLUT3 protein, the main glucose transporters in the brain, were decreased in the cerebral cortex of both SAMR1 and SAMP8 mice when compared with the corresponding 4-month age animals [82].”.
And see Line 362: “Glutamic acid decarboxylase (EC 4.1.1.15; GAD) promotes the transformation of excessive glutamate to γ-aminobutyric acid (GABA). Interestingly, GAD activity was lower in cortex from older (24 months) C57BL/6J mice compared to younger mice (4 months) [88]. The lower GAD activity could be a main contributor to the higher glutamate content in 24 months old compared to 4 months old C57BL/6J mice that we reported here, as we found no changes on EAAT2 protein levels (i.e. glutamate uptake) between 4 and 24 months.”.
However, the idea that reduced glutamate uptake may be responsible for increasing glutamate levels in the cortical tissue is strange. The authors mean reduce vesicular glutamate uptake? The reduced VGluT1 cited from [64] may be related to a reduction in glutamatergic terminals. And, the authors must consider that the main responsible for glutamate uptake are the cell membrane EAATs.
As mentioned above, we have rewritten this part concerning glutamate levels, and the possible role of VGLUTs has been clarified. In addition, we have measured EAAT2 protein expression levels in our samples. These new results have been included in the new Figure 9. See Line 335: “Regarding glutamate uptake, the neuroprotective capacity of astrocytes by clearing extracellular glutamate has been analyzed in primary cultures enriched in astrocytes using cerebral cortical tissue from 2‐day‐old SAMP8 and SAMR1 mice. Interestingly, glutamate uptake in SAMP8 astrocyte cultures was significantly lesser than in SAMR1 cultures, while the expression of the glutamate‐aspartate transporter (GLAST) was not different between SAMP8 and SAMR1 astrocytes. Moreover, the neuroprotective capacity of astrocytes, evaluated in co‐cultures with cortical neurons, was lower in SAMP8 astrocytes [83]. In addition, astrocytes aged in vitro have a reduced ability to maintain neuronal survival [84]. This reduction in glial glutamate uptake capacity in SAMP8 may elevate extracellular glutamate levels leading to neuronal excitotoxicity [85], and contribute to the premature learning and memory deficits observed in this murine model of early aging but also of AD. In fact, astrocyte senescence and its putative role in the pathologic progress of AD has been recently reviewed [86]. Astrocytes cultured from neonatal SAMP8 mice present similar alterations to those described in the whole brains of SAMP8 mice at 1–5 months of age [83]. Thus, it is conceivable that astrocytes (mal)function could contribute to the significantly higher glutamate levels that we detected in 12 months old SAMP8 mice when compared with SAMR1 strain.
In line with this, we detected an increase in EAAT2 levels in SAMR1 mice from 6 to 12 months, when glutamate levels are lower. Furthermore, in SAMP8 mice the highest EAAT2 protein level is detected at 6 months, when glutamate content is lowest, suggesting the importance of glutamate uptake in the final glutamate level found in the cerebral cortex of SAM strains.
Another contributing factor to glutamate levels could be a reduced glutamate uptake into synaptic vesicles. In fact, protein expression of vesicular glutamate transporter (VGLUT) isoforms 1, 2 and 3, and synaptophysin, was decreased in an age-dependent manner in the cerebral cortex of SAMP8 mice of 2, 6 and 12 months, which could indicate that the glutamatergic synaptic transmission was weakened in the brain of aging SAMP8 [87].”.
-
In the paragraph regarding adenosine receptors, again it is not clear whether the reduced expression of A1 and increased of A2A were observed at the same time points from the present study. And, the last phrase is puzzling: what is the relationship among 5´-NT and A2AR interaction and glutamate release? The suggestion that A2AR may have a role on regulating glutamate uptake and release it is known (Ciruela et al., 2006; Matos et al., 2013; 2015). This discussion should be reorganized.
We realize that these paragraphs must be explained better. We have included these references and others to discuss the possible role of 5’-NT/A2A/Glutamate relationship in our results. The final sentences are (see Line 403): “Moreover, overfunction of A2A receptors accelerates neurodegeneration process [14,42], and the apparently preserved levels of A2A receptors in SAMP8 mice during aging would assure some pathological events, for instance, an overstimulated glutamate release [10,99]. Furthermore, a key regulator of A2A receptor-mediated signaling seems to be 5’NT (reviewed in [100]). The co-location and physical association of 5’NT and A2A receptors has been reported within same membrane microdomains, for instance, in synaptosomes from striatum of C57BL/6 animals [101,102]. Thus, extracellularly generated adenosine would immediately activate A2A receptors, which in turn would promote glutamate release. Interestingly, temporal profile of glutamate content in SAMP8 and C57BL/6J mice reported here was paralleled to plasma 5’-NT activity during aging.”.
Methods:
-
It is not clear the exact number of mice used to each experiment. Please clarify the N to each group.
We have stated the N value for each experimental age group in the Methods section as follows (see Line 415): “Cerebral cortex from 33 male mice were used in this work: 13 SAMR1 (3 months, n=5; 6 months, n=5; 12 months, n=3), 9 SAMP8 (3 months, n=3; 6 months, n=3; 12 months, n=3), and 11 C57BL/6J (4 months, n=7; 24 months, n=4) mice.”.
Conclusions:
-
The conclusions are also not very And the following phrase is not correct: “Notably, 5’NT activity similarly behaved as glutamate levels in SAMP8 mice … “
We have rewritten the conclusions to avoid misinterpretations, and the phrase you mention has been deleted.
It seems the authors are considering that 5´-NT activity decrease in 12 months would be responsible for increasing glutamate levels? It seems to me that the relationship between 5´- NT and A2aR localization and interaction (proteins expression) may not be explained by 5´-NT activity.
Again, we agree this paragraph needs to be improved. In fact, we have not reported a decrease but an increase in 5’-NT activity at 12 months in SAMP8 mice, as Figure 1 shows. We suggest a parallelism between plasma membrane 5’NT activity and glutamate levels through A2A receptors activation during aging in SAMP8 mice, which could be also valid to C57BL/6J mice.
We have clarified this point as follows (see Line 407): “The co-location and physical association of 5’NT and A2A receptors has been reported within same membrane microdomains, for instance, in synaptosomes from striatum of C57BL/6 animals [101,102]. Thus, extracellularly generated adenosine would immediately activate A2A receptors, which in turn would promote glutamate release. Interestingly, temporal profile of glutamate content in SAMP8 and C57BL/6J mice reported here was paralleled to plasma 5’-NT activity during aging.”.

Reviewer 2 Report
In their work Sanchez-Melgar and coworkers analyzed several purinergic parameters (activity of 5’NT, adenosine and its related metabolites, levels of AMPK and p-AMPK) and glutamate levels in the cerebral cortex of three strains of mice (SAMR1 – resistant to senescence, SAMP8 – Senescence Accelerated Mouse Prone, and C57BL/6J). Experiments were performed on mice at different ages: SAMR1 and SAMP8 - 3, 6 and 12 months, while C57BL/6J - 4 and 24 months. 5’NT activity was lower in 6 months old SAMP8 mice compared to animals at the age of 3 and 12 months. On the contrary, the enzyme activity in the cerebral cortex of 6 months old SAMR1 mice was higher than those found in animals aged 3 and 12 months. No age-dependent changes were observed in C57BL/6J. Levels of xanthine and guanosine in the cerebral cortex of 6 months old SAMR1 were significantly higher compared to younger and older animals. Glutamate levels in tissue samples collected from 12 months old SAMR1 mice were markedly lower compared to younger animals, whereas in SAMP8 mice they were significantly higher than in younger animals. It is suggested that adenosinergic metabolism in involved in the neurodegenerative processes underlying Alzheimer disease.
Comments:
- It is totally unclear why in these comparative studies the Authors used C57BL/6J mice at the age of 4 and 24 months and the other two strains at different ages (3, 6 and 12 months). The rationale underlying such a paradigm must be explained.
- In the majority of the studied parameters, results obtained for the 6 months old mice markedly differ from these found in younger and older animals. The Authors should at least try to discuss which processes could possible underlay these phenomena. Just for example, why altered 5’NT activity did normalize between 6 and 12 months?
- It is unclear how many mice for each age group were used. This information should be given.
- I strongly suggest to delete a fragment of discussion on age-dependent changes in 5’NT positive subpopulation of B cells (lines 188-194).
- The Discussion is not well focused and balanced.
Author Response
Reviewer: 2
In their work Sanchez-Melgar and coworkers analyzed several purinergic parameters (activity of 5’NT, adenosine and its related metabolites, levels of AMPK and p-AMPK) and glutamate levels in the cerebral cortex of three strains of mice (SAMR1 – resistant to senescence, SAMP8
– Senescence Accelerated Mouse Prone, and C57BL/6J). Experiments were performed on mice at different ages: SAMR1 and SAMP8 - 3, 6 and 12 months, while C57BL/6J - 4 and 24 months. 5’NT activity was lower in 6 months old SAMP8 mice compared to animals at the age of 3 and 12 months. On the contrary, the enzyme activity in the cerebral cortex of 6 months old SAMR1 mice was higher than those found in animals aged 3 and 12 months. No age-dependent changes were observed in C57BL/6J. Levels of xanthine and guanosine in the cerebral cortex of 6 months old SAMR1 were significantly higher compared to younger and older animals. Glutamate levels in tissue samples collected from 12 months old SAMR1 mice were markedly lower compared to younger animals, whereas in SAMP8 mice they were significantly higher than in younger animals. It is suggested that adenosinergic metabolism in involved in the neurodegenerative processes underlying Alzheimer disease.
Comments:
It is totally unclear why in these comparative studies the Authors used C57BL/6J mice at the age of 4 and 24 months and the other two strains at different ages (3, 6 and 12 months). The rationale underlying such a paradigm must be explained.
We have included in the Introduction section the rationale for selecting these mice strains of the analyzed ages. See Line 67: “Mice are extensively used in aging and senescence research due to their genetic and physiological similarity to humans [24]. Wild-type C57BL/6J is the most widely used inbred strain and they are frequently used in studies related to aging and neurodegenerative disorders [25-27]. Senescence-accelerated mouse-prone 8 (SAMP8) is an ideal model to study AD, displaying age-related learning and memory disorders [28-32], and senescence-accelerated mouse-resistant 1 (SAMR1) is usually considered as the control strain for SAMP8 mice studies, with also accelerated senescence but without learning and memory impairments [28,29].
The relationship between first occurred AD-like changes and monthly age in the brain of SAMP8 mice has been reported [28]. In brief, at 3 months of age, these mice have aberrant gene expression, oxidative stress and Tau hyperphosphorylation. At 6 months, they develop cognitive impairment, glial degeneration, inflammation, and Aβ deposition. Finally, at 12 months of age, they show alterations in dendrites, synapses, neurons, and Aβ plaques and neuron loss. The lifespan of SAMP8 mice is about 10–12 months, while SAMR1 mice is about 19–21 months [29,33]. Regarding C57BL/6J, its lifespan is about 26–28 months [34,35]. As C57BL/6J mice exhibit a longer lifespan than SAMR1 and SAMP8 mice, we selected 4 and 24 months of age, approximately at the same survival point as seen for 3 and 12 month-old SAM mice [33,36]. In fact, it has been reported that SAMP8 mice at 1-year-old showed age-associated characteristics in tissue histology, proliferation and differentiation, which were comparable to 2-year-old mice of the C57BL/6 strain [37].”
In the majority of the studied parameters, results obtained for the 6 months old mice markedly differ from these found in younger and older animals. The Authors should at least try to discuss which processes could possible underlay these phenomena. Just for example, why altered 5’NT activity did normalize between 6 and 12 months?
The recovery of 5’NT activity at 12 months is an interesting issue to which we have not a direct and easy explanation. If we compare the percentage of change versus the corresponding value at 3 months, it seems that a tight relationship exists between 5’NT activity, EAAT2, AMPK and Glutamate levels in SAMP8 mice, which is not seen in SAMR1.
Our results highlight that age is a critical modifier of 5’NT and others parameters. However, with the exception of reference [15], we have not found in the literature similar studies analyzing 5’NT activity in, at least, three ages covering the lifespan of these animal models. Further investigation would be required to reveal the precise molecular mechanism behind this biphasic response in SAMP8 mice.
In the Discussion section, we have included the sentences (see Line 293): “Regarding SAM strains, the activity of AMPK in the cerebral cortex of young (2-month-old) SAMP8 mice was reported significantly higher than that of SAMR1 controls, thus inhibiting tau hyperphosphorylation [73]. These authors suggested that AMPK activation at an early stage (i.e. presymptomatic 2-month-old SAMP8 mice) without prominent AD pathologies plays a protective role against detrimental stress in the cerebral cortex, as activation of cortical AMPK inhibits the GSK3b-mediated hyperphosphorylation of tau. In addition, activated AMPK (p- AMPK) accumulated in pre-tangle- and tangle-bearing neurons in postmortem brain samples in AD and other major tauopathies [74]. We have not directly compared the levels and activity of AMPK between SAMP8 and SAMR1 mice, as these samples were electrophoresed in different gels. However, it is interesting to note that at the age of 6 months total AMPK levels significantly decrease in SAMP8 mice, and the most differences (e.g. 5’NT, guanosine, hypoxanthine, glutamate) between these two SAM strains are detected at this age. This would point to a loss of the cited AMPK-mediated compensatory mechanism at the age of 6 months. Age‐related changes in the functioning of AMPK have been reviewed both at the cell and organism levels. Unfortunately, further investigation is needed to accurately determine how aging modulates the activity and expression of AMPK, mainly in mammals [46].”
See in the Discussion section (Lines 309-367) several paragraphs concerning glutamate levels. Line 359: “Therefore, glutamate synthesis from glucose can be the main, but not unique, contributing factor to glutamate levels at 6 months of age, while glutamate uptake processes would be more important at 12 months in both SAMR1 and SAMP8 mice.”
In addition, we stated in Line 376: “All together, these results might point out 6 months- old SAMP8 mice as the most affected among the three strains analyzed in this work. SAMP8 mice constitute both a senescence-accelerated model and a model for studying the initial neurodegenerative alterations in AD [30-32,91,92]. In fact, impairments in spatial learning and increased oxidative stress can be detected in SAMP8 mice as early as 3 months old, followed by the impairment in spatial memory and increased tau hyperphosphorylation at 5 months, hippocampal cognitive deficits at 6 months, and gliosis and increased levels of soluble Aβ at 8 months [28,31]”.
It is unclear how many mice for each age group were used. This information should be given.
We have stated the N value for each experimental age group in the Methods section as follows (see Line 415): “Cerebral cortex from 33 male mice were used in this work: 13 SAMR1 (3 months, n=5; 6 months, n=5; 12 months, n=3), 9 SAMP8 (3 months, n=3; 6 months, n=3; 12 months, n=3), and 11 C57BL/6J (4 months, n=7; 24 months, n=4) mice.”.
I strongly suggest to delete a fragment of discussion on age-dependent changes in 5’NT positive subpopulation of B cells (lines 188-194).
We have deleted these sentences as we realize they are not important for the discussion.
The Discussion is not well focused and balanced.
We have rewritten several paragraphs of the Discussion section, and we hope now is more complete and balanced.

Reviewer 3 Report
The manuscript by Sanchez-Melgar is interesting and provides information on the neuromodulator Adenosine and its metabolites and the neurotransmitter glutamate and its receptor AMPA in the cerebral cortex of strains of mice with or without a phenotype for Alzheimer's disease (AD).
There are several major and minor caveats and the authors need to fix these issues.
Abstract:
Need to indicate AD is Alzheimer's disease.
Need to introduce the strains used in the beginning of the abstract instead of the end (only the last sentence currently states that SAMP8 is model of aging and AD). Also need to define SAMP8 and SAMR1
Introduction:
Needs a good rationale for using the strains of mice and the age of mice selected. Also needs a rationale for examining the cerebral cortex. None of this information is provided and it is not clear why the authors performed the study.
Results:
For figure 1,3 and figure 6, the authors have incorrectly performed the statistical analysis. The figure represents a two-way design with age and stain as independent variables. However a one-way ANOVA was performed. This is incorrect. Please re-do the analysis and report it correctly.
The authors have used 3 ages of SAM mice (3, 6 and 12 months). However, they have used 2 age groups of C57 mice (4 and 24 months). It is not clear why they have such discrepancy with the age between the stains. What were they trying to compare here? This needs to be justified. An additional group of C57 at 12 months will be required to compare any data they have so far with the SAM mice.
It will be necessary to measure ADA activity in the brain samples they have collected to reveal the role of ADA in the differential activity of 5’NT in the strains and different ages.
Discussion:
How does the age of C57 compare with SAM mice used in the study? Why was C57 used and what does the data from this strain provide?
The metabolites were analyzed. However, they have not been discussed in detail. In addition to inosine what do the other metabolites reveal?
Rewrite the sentence ‘However, A2A receptor binding was almost duplicated…’ Lines 267-268…. The sentence is confusing.
Author Response
Reviewer: 4
The manuscript by Sanchez-Melgar is interesting and provides information on the neuromodulator Adenosine and its metabolites and the neurotransmitter glutamate and its receptor AMPA in the cerebral cortex of strains of mice with or without a phenotype for Alzheimer's disease (AD).
There are several major and minor caveats and the authors need to fix these issues. Abstract:
Need to indicate AD is Alzheimer's disease.
Need to introduce the strains used in the beginning of the abstract instead of the end (only the last sentence currently states that SAMP8 is model of aging and AD). Also need to define SAMP8 and SAMR1
We have modified the Abstract to define the strains analyzed, and to include new results regarding ADA activity and EAAT2 levels.
Introduction:
Needs a good rationale for using the strains of mice and the age of mice selected. Also needs a rationale for examining the cerebral cortex. None of this information is provided and it is not clear why the authors performed the study.
We have included in the Introduction section the rationale for selecting these mice strains of the analyzed ages. See Line 67: “Mice are extensively used in aging and senescence research due to their genetic and physiological similarity to humans [24]. Wild-type C57BL/6J is the most widely used inbred strain and they are frequently used in studies related to aging and neurodegenerative disorders [25-27]. Senescence-accelerated mouse-prone 8 (SAMP8) is an ideal model to study AD, displaying age-related learning and memory disorders [28-32], and senescence-accelerated mouse-resistant 1 (SAMR1) is usually considered as the control strain for SAMP8 mice studies, with also accelerated senescence but without learning and memory impairments [28,29].
The relationship between first occurred AD-like changes and monthly age in the brain of SAMP8 mice has been reported [28]. In brief, at 3 months of age, these mice have aberrant gene expression, oxidative stress and Tau hyperphosphorylation. At 6 months, they develop cognitive impairment, glial degeneration, inflammation, and Aβ deposition. Finally, at 12 months of age, they show alterations in dendrites, synapses, neurons, and Aβ plaques and neuron loss. The lifespan of SAMP8 mice is about 10–12 months, while SAMR1 mice is about 19–21 months [29,33]. Regarding C57BL/6J, its lifespan is about 26–28 months [34,35]. As C57BL/6J mice exhibit a longer lifespan than SAMR1 and SAMP8 mice, we selected 4 and 24 months of age, approximately at the same survival point as seen for 3 and 12 month-old SAM mice [33,36]. In fact, it has been reported that SAMP8 mice at 1-year-old showed age-associated characteristics in tissue histology, proliferation and differentiation, which were comparable to 2-year-old mice of the C57BL/6 strain [37].”
The rationale for studying cerebral cortex has been included (see Line 55): “AD is characterized by the extracellular deposition of amyloid β-peptide (Aβ) plaques and the intracellular formation of neurofibrillary tangles in the cerebral cortex [18]. In addition, age affects the brain independently of AD [19], and there is a growing number of proponents of the view that it is important to understand age-related changes to better understand AD. Therefore, studying the brain regions that are vulnerable to both aging and AD, as cerebral cortex, may allow us to disentangle disease-specific mechanisms from normal age-related events [20,21].”.
Results:
For figure 1,3 and figure 6, the authors have incorrectly performed the statistical analysis. The figure represents a two-way design with age and stain as independent variables. However a one-way ANOVA was performed. This is incorrect. Please re-do the analysis and report it correctly.
We made a mistake in the figure legend description, as we indeed performed a two-way ANOVA analysis but stated “One-way” instead of “Two-way”. Perhaps because of copy /paste text from Figure 5 legend, where “One-way” was correctly performed and stated. We are sorry for that. Nevertheless, some missing asterisks have been included in Figure 1. In summary, “One-way” was performed and stated in figures 1, 3, 4 and 7, and “Two-way” in figures 6 and 9.
The authors have used 3 ages of SAM mice (3, 6 and 12 months). However, they have used 2 age groups of C57 mice (4 and 24 months). It is not clear why they have such discrepancy with the age between the stains. What were they trying to compare here? This needs to be justified. An additional group of C57 at 12 months will be required to compare any data they have so far with the SAM mice.
Please, see the answer given above about the rationale for choosing these strains and ages (see Line 67).
We are sorry, but we have not available an experimental group of 12-month-old C57BL/6J mice.
It will be necessary to measure ADA activity in the brain samples they have collected to reveal the role of ADA in the differential activity of 5’NT in the strains and different ages.
According to your suggestion, we have measured ADA activity and a new Figure 3 has been included in the revised version of the manuscript (see Line 55).
Discussion:
How does the age of C57 compare with SAM mice used in the study? Why was C57 used and what does the data from this strain provide?
Please, see the answer given above about the rationale for choosing these strains and ages (see Line 67).
The metabolites were analyzed. However, they have not been discussed in detail. In addition to inosine what do the other metabolites reveal?
We have included the relationship between guanosine a 5’-nucleotidase, which is particularly interesting in SAMR1 mice. Therefore, we have included the following paragraph (see Line 262): “Interestingly, detected 5’NT activities (plasma membrane and cytosolic) and guanosine levels were lower in SAMP8 than in SAMR1 mice. Moreover, the increased guanosine levels detected in 6 months versus 3 months SAMR1 mice could be, at least in part, a reflect of the also increased 5’NT activity measured at this age. However, this positive relationship between 5’NT and guanosine values is apparently lost in SAMP8 mice.”
Rewrite the sentence ‘However, A2A receptor binding was almost duplicated…’ Lines 267- 268…. The sentence is confusing.
We have rewritten this sentence as (see Line 391): “Thus, A1 receptor binding to cerebral cortex in old (24 months) Wistar rats was a 40 % of detected in young (6 weeks) rats, while A2A receptor binding was two-fold higher in old than in young rats [41].”.

Round 2
Reviewer 2 Report
No further comments and suggestions for Authors.
Author Response
Thank you for your evaluation of our work.
Reviewer 3 Report
The authors have mostly addressed the concerns indicated by this reviewer. However, some minor issues remain.
The authors need to provide details about their statistical analysis. As indicated in the previous review, the authors need to discuss their two-way results. Please indicate the interaction, the main effects, etc. Please do this for all the results. Just indicating a significance is not sufficient.
Author Response
The authors have mostly addressed the concerns indicated by this reviewer. However, some minor issues remain.
The authors need to provide details about their statistical analysis. As indicated in the previous review, the authors need to discuss their two-way results. Please indicate the interaction, the main effects, etc. Please do this for all the results. Just indicating a significance is not sufficient.
We have included in the Methods section (line 528) the sentences “Overall comparisons between the SAMR1 and SAMP8 mice were tested with two-way ANOVA. The main effects tested were, strain (measured parameter differs between the SAMR1 and SAMP8 mice), age (measured parameter changes with age), and interaction between strain and age (age-related changes are different between the SAMR1 and SAMP8 mice).”.
In the Results section, we have also included all required details about the two way-ANOVA analysis performed in Figures 1, 3, 4 and 7.
In line 132: “In plasma membrane fraction, two-way ANOVA (strain x age) revealed a significant interaction between factors (F(2, 15) = 9.360, P=0.0023) and a non-significant main effect of strain (F(1, 15) = 0.6574, P=0.4302) and a non-significant effect of age (F(2, 15) = 1.248, P=0.3154). In cytosolic fraction, two-way ANOVA revealed a significant interaction between factors (F(2, 14) = 5.461, P=0.0176) and a significant main effect of strain (F(1, 14) = 12.29, P=0.0035) and a non-significant effect of age (F(2, 14) = 0.09649, P=0.9086).”.
In line 158: “Two-way ANOVA (strain x age) revealed a significant main effect of age (F(2, 13) = 6.821, P=0.0094), and a non-significant main effect of strain (F(1, 13) = 0.02968, P=0.8659) and a non-significant interaction between factors (F(2, 13) = 1.403, P=0.2808).”.
In line 179: “Two-way ANOVA revealed no significant changes in adenosine levels with age (F(2, 15) = 0.6746, P=0.5242), strain (F(1, 15) = 1.250, P=0.2811) or age × strain (F(2, 15) =
0.3222, P=0.7294). For inosine levels, two-way ANOVA revealed no significant changes with age (F(2, 14) = 0.5776, P=0.5741) and age × strain (F(2, 14) = 0.4688, P=0.6352), and significant changes with strain (F(1, 14) = 5.423, P=0.0354). For xanthine levels, two-way ANOVA revealed no significant changes with strain (F(1, 15) = 0.1877, P=0.6710) and age × strain (F(2, 15) = 0.2410, P=0.7889), and significant changes with age (F(2, 15) = 3.916, P=0.0428). For hypoxanthine levels, two-way ANOVA revealed no significant changes with strain (F(1, 14) = 2.974, P=0.1066), age (F(2, 14) = 0.03323, P=0.9674), and age × strain (F(2, 14) = 3.698, P=0.0514). For guanosine levels, two-way ANOVA revealed no significant changes with strain (F(1, 14) = 2.334, P=0.1488) and age × strain (F(2, 14) = 2.264, P=0.1406), and significant changes with age (F(2, 14) = 5.752, P=0.0150).”.
In line 229: “Two-way ANOVA (strain x age) revealed a non-significant main effect of age (F(2, 15) = 1.016, P=0.3857) and strain (F(1, 15) = 0.1543, P=0.7000), and a significant interaction between factors (F(2, 15) = 7.357, P=0.0059).”.
